# Cervical pessary versus vaginal progesterone in women with a multiple pregnancy and a short cervix: A randomised controlled trial

Charlotte E. van Dijk[1,2]*, Annabelle L. van Gils[1,2], Maud D. van Zijl[1,2], Bouchra Koullali[1,2], Marijke C. van der Weide[1], Eline S. van den Akker[3], Brenda J. Hermsen[3], Joris van Drongelen[4], Marjon A. de Boer[2,5], Wilhelmina M. van Baal[6], Karlijn C. Vollebregt[7], Flip W. van der Made[8], Sanne J. Gordijn[9], Marieke Sueters[10], Lia D. E. Wijnberger[11], Martijn A. Oudijk[2,5], Ben W. J. Mol[1,12], Brenda M. Kazemier[1,2], Eva Pajkrt[1,2], for the Quadruple P Research Group¶

1 Department of Obstetrics and Gynaecology, Amsterdam UMC, University of Amsterdam, Amsterdam, the Netherlands, 2 Amsterdam Reproduction and Development Research Institute, Amsterdam, the Netherlands, 3 Department of Obstetrics and Gynaecology, Onze Lieve Vrouwe Gasthuis, Amsterdam, the Netherlands, 4 Department of Obstetrics and Gynaecology, Radboud University Medical Centre, Radboud University, Nijmegen, the Netherlands, 5 Department of Obstetrics and Gynaecology, Amsterdam UMC, Vrije Universiteit Amsterdam, Amsterdam, the Netherlands, 6 Department of Obstetrics and Gynaecology, Flevoziekenhuis, Almere, the Netherlands, 7 Department of Obstetrics and Gynaecology, Spaarne Gasthuis, Haarlem, the Netherlands, 8 Department of Obstetrics and Gynaecology, Sint Fransiscus Gasthuis, Rotterdam, the Netherlands, 9 Department of Obstetrics, University Medical Centre Groningen, University of Groningen, Groningen, the Netherlands, 10 Department of Obstetrics and Gynaecology, Leiden University Medical Centre, University of Leiden, Leiden, the Netherlands, 11 Department of Obstetrics and Gynaecology, Rijnstate Ziekenhuis, Arnhem, the Netherlands, 12 Department of Obstetrics and Gynaecology, Monash University, Clayton, Victoria, Australia

¶ Membership of Quadruple P Research group is provided in the Acknowledgments.
* c.e.vandijk@amsterdamumc.nl

## Abstract

### Background

In absence of direct comparisons, consensus on the preferred preventive treatment for multiple pregnancies with a short cervix is lacking. Therefore, we compared the effectiveness of a cervical pessary and vaginal progesterone in the prevention of adverse perinatal outcomes and preterm birth (PTB) in women with a multiple pregnancy, no prior spontaneous PTB (sPTB) before 34 weeks' gestation, and an asymptomatic mid-trimester shortened cervix below 38 mm.

### Methods and findings

This open-label, superiority, multi-centre randomised controlled trial was conducted in 20 hospitals in the Netherlands. Women with a healthy multiple pregnancy and an asymptomatic cervical length (CL) below 38 mm between 16 and 22 weeks' gestation were eligible, with a target sample size of 332. Following an interim analysis, the study was halted for futility. A total of 276 multiples, including seven triplet

**Data availability statement:** The de-identified individual participant data (IPD) that underlie the results reported in this article cannot be made publicly available because of ethical and legal restrictions related to patient privacy. Data requests for scientifically sound projects may be submitted to the Office of Clinical Research in Obstetrics and Gynecology which is independent of the study authors, at onder-zoeksbureauvkc@amsterdamumc.nl. Requests will be evaluated in accordance with applicable ethical approvals and data use agreements. All aggregate data supporting the findings are provided within the article and its Supporting information files.

**Funding:** This study was supported by Stichting Stoptevroegbevallen (https://www.stoptevroeg-bevallen.nl), a non-profit research foundation supporting scientific projects in the field of preterm birth. EP and MAO are both board members at Stichting Stoptevroegbevallen. The funder itself had no other role in study design, data collection and analysis, decision to publish, or preparation of the manuscript.

**Competing interests:** BWJM reports consultancy for Merck KGaA, Organon, and Norgine, and travel support from Merck KGaA. He has received an NHMRC Investigator Grant (GNT1176437) and holds stock in ObsEva. EP reports institutional support from Stichting Stop Te Vroeg Bevallen, Amsterdam Research and Development, and ZonMw (various grants). She received honoraria paid to the foundation for webinars/symposia, served as Medical Advisor for Stichting Prenatale Screening Noord Holland (payment to institution), chaired the Review Committee on late pregnancy termination and neonatal end-of-life decisions (appointed by the Dutch Ministry of Health/Justice), and is board member of Stichting Stop Te Vroeg Bevallen (expenses reimbursed, unpaid). SJG reports institutional grants from ZonMw (CEPRA study, intrapartum pain relief stimulation grant, Drigitat study), a private fund ("Cycling for Frederik"), and subcontracting within an ERC project (FGR PODS), as well as payments to the institution from the SCEM symposium/webinar and CVOI. She has received kits free of charge from Roche and Thermofisher for unrelated studies. She served on the International Stillbirth Alliance board

pregnancies, were randomised 1:1 to receive either an Arabin cervical pessary (N = 138) or vaginal progesterone 200 mg daily (N = 138) until 36 weeks' gestation or earlier if indicated. The primary outcome was a composite adverse perinatal outcome, with secondary outcomes including rates of (s)PTB before 24, 28, 32, 34, and 37 weeks. Predefined subgroup analyses were conducted based on CL, parity, chorionicity, and number of foetuses. Among 531 neonates (pessary N = 269, progesterone N = 262), the composite adverse perinatal outcome occurred in 19.7% of neonates in the pessary group versus 13.7% in the progesterone group (crude RR 1.43; 95% CI [0.85,2.4], $p = 0.18$). The rates of (s)PTB were not significantly different between groups. In the subgroup with a CL of ≤25 mm, no significant difference in the composite perinatal outcome was found (41.1% versus 34.7%, RR 1.18; 95% CI [0.60,2.33], interaction $p = 0.63$). However, among nulliparous women, the composite outcome was more frequent in the pessary group compared to progesterone (30.0% versus 15.9%, RR 1.88; 95% CI [1.03,3.43], interaction $p = 0.93$). The study's main limitations include the inability to blind interventions, potentially introducing bias, and low self-reported medication compliance in the progesterone group, which may have led to overestimated adherence and underestimated progesterone's preventive potential in the per-protocol analysis.

## Conclusion

In women with multiple pregnancies and a midtrimester short cervix below 38 mm, we found no superiority of a cervical pessary compared to vaginal progesterone the prevention of perinatal complications. While progesterone may have a modest effect, future studies should focus on other interventions in multiple pregnancies such as a cerclage, both ultrasound- and physical examination-indicated.

## Trial registration

This trial was registered at the International Clinical Trial Registry Platform (ICTRP, EUCTR2013-002884-24-NL, https://trialsearch.who.int/Trial2.aspx?TrialID=EUCTR2013-002884-24-NL).

## Author summary

### Why was this study done?

- Women with multiple pregnancies are at higher risk of giving birth too early, even more when they have a short cervix, which can lead to serious health problems for their babies.

- Two treatments, vaginal progesterone and a cervical pessary, are often used to try to prevent early birth, but it's unclear which one works better.

(2017–2022), scientific committees of the Dutch Society of Obstetrics and Gynaecology, and chaired its placental insufficiency special interest group. MAO reports institutional grants from ZonMw (TWINC study) and Amsterdam UMC (cervical length measurement project), and a grant obtained by Pregnolia for cervical stiffness research. He holds leadership roles as chairman of the Fetal-Maternal Medicine board and scientific committee of the Dutch Society of Obstetrics and Gynaecology, and is board member of Stichting Stop Te Vroeg Bevallen (expenses reimbursed, unpaid). CED reports a WPMZ travel grant for a presentation on another study. All other authors, ALG, MDZ, BK, MCW, ESA, BJH, JD, MAB, WMB, KCV, FWM, MS, LDEW, en BMK, declare that no competing interests exist.

**Abbreviations:** BPD, Bronchopulmonary Dysplasia; CI, confidence interval; CL, cervical length; CONSORT, Consolidated Standards of Reporting Trials; CROWN, Core Outcomes in Women's Health; DSMC, Data Safety Monitoring Committee; GA, gestational age; GEE, generalised estimating equations; HELLP, haemolysis, elevated liver enzymes and low platelets; IDR, Inter Quartal Range; ISCI, intra-cytoplasmic sperm injections; ITT, intention to treat; IVF, in vitro fertilisation; IVH, intraventricular haemorrhage; LLETZ, large loop excision of the transformation zone; NEC, necrotising enterocolitis; NICU, neonatal intensive care unit; PDC, proportion of days covered; PP, per protocol; PROM, premature rupture of membranes; PTB, preterm birth; PVL, periventricular leukomalacia; RCTs, randomised clinical trials; RDS, Severe respiratory distress syndrome; ROP, retinopathy of prematurity; RRs, relative risks; SD, standard deviation; sPTB, spontaneous preterm birth.

- We aimed to investigate if a cervical pessary is more effective than vaginal progesterone for women with a multiple pregnancy and a short cervix.

## What did the researchers do and find?

- We conducted a study in 20 hospitals in the Netherlands where 276 women with multiple pregnancies and a short cervix were randomly assigned to receive either a vaginal pessary or daily vaginal progesterone. The study was halted for futility.

- We found no clear advantage of using a pessary compared to progesterone in preventing serious health problems in newborns or in reducing the chance of being born too early.

- In first-time mothers, the pessary group actually had worse outcomes than the progesterone group, with more babies affected by complications.

## What do these findings mean?

- This study suggests that a cervical pessary should not be used instead of progesterone to prevent early birth in women with a multiple pregnancy and a short cervix.

- Vaginal progesterone might still offer some benefit, especially for women having their first baby, but more research (meta-analysis) is needed to confirm this.

- The study's main limitations include the inability to blind interventions, potentially introducing bias, and low self-reported medication compliance in the progesterone group, which may have led to overestimated adherence and underestimated progesterone's preventive potential in the per-protocol analysis.

## Introduction

The primary challenge in multiple pregnancies is the prevention of extremely or very preterm birth (PTB), defined as delivery before 28 and 32 weeks' gestation, respectively, as approximately 10% results in delivery before 32 weeks' gestation. Up to 40% will deliver spontaneously before 37 weeks' gestation [1]. Early PTB in multiple pregnancies considerably contributes to increased perinatal mortality, neonatal morbidity, and long-term health issues in the offspring born from these pregnancies [2,3]. PTB associated adverse outcomes are more common in women with multiple pregnancies compared to women with singleton pregnancies [4,5]. For example, infant mortality rates are substantially higher, with 21 deaths per 1,000 live births in twins versus 5 per 1,000 in singletons [6,7]. The higher rate of spontaneous preterm birth (sPTB) in twin pregnancies is partly attributed to greater myometrial stretching, contributing to more frequent and intense myometrial contractility compared to singleton pregnancies [8,9]. Another important risk indicator for sPTB is a short cervical length (CL). In women with a singleton or a multiple pregnancy, progesterone has been

demonstrated to effectively reduce PTB in cases of short cervix, with cut-offs suggested at 25 or 30 mm for singletons and multiples [10,11].

Three randomised clinical trials (RCTs) in women with multiple pregnancy with a short cervix (defined between <25 and <38mm) found pessary treatment to reduce sPTB rates of before 34 weeks' gestation, therefore, a cervical pessary has been suggested as a low-cost, effective intervention for the prevention of PTB [12–14]. However, other RCTs could not confirm these outcomes [15–18]. In many RCTs, participants in both the pessary and non-pessary groups also received progesterone, thus hampering a direct comparison. In absence of direct comparisons, consensus on the preferred preventive treatment for multiples with a short cervix is lacking. Therefore, we compared the effectiveness of a cervical pessary and vaginal progesterone in the prevention of sPTB and adverse perinatal outcomes in women with a multiple pregnancy, no prior sPTB before 34 weeks' gestation, and an asymptomatic mid-trimester shortened cervix.

## Methods

### Ethical approval

Ethical approval was obtained from the Medical Research Ethics Committee of the Amsterdam University Medical Centre (MEC AMC 2013_019) while the boards of all participating centres approved local execution (S4 File: Study Approval).

### Study design and participants

The Quadruple P study (Pessary or Progesterone to Prevent Preterm delivery in pregnant women with a short CL) is a multicentre, open-label, randomised controlled trial with a superiority design, comparing the effectiveness of cervical pessary and vaginal progesterone. Ethical approval was granted by the Medical Research Ethics Committee of the Amsterdam University Medical Centre (MEC AMC 2013_019; 19th November 2013) (S4 File: Study Approval), and all participating centres approved local execution. The trial was registered at the International Clinical Trial Registry Platform (ICTRP, EUCTR2013-002884-24-NL, https://trialsearch.who.int/Trial2.aspx?TrialID=EUCTR2013-002884-24-NL.).

The protocol of this study was written for both women with a singleton and women with a multiple pregnancy, yet analyses and publication were planned separately (S2 File: Study Protocol) [19]. Where the words women, she and her are used, it is to describe individuals whose sex was assigned as birth as female, whether they identify as female, male or binary. The results for women with singleton pregnancies have already been published [20]. The study was carried out in 20 hospitals in the Netherlands, within the Dutch Consortium for Healthcare Evaluation and Research in Obstetrics and Gynaecology (NVOG Consortium). This study adheres to the Consolidated Standards of Reporting Trials (CONSORT) guidelines (S1 CONSORT Checklist) [21]. An independent Data Safety Monitoring Committee (DSMC) provided oversight. Details of all protocol amendments, including dates of approval by the institutional ethics committee, as well as any deviations from the study protocol, are provided in Table A in S1 File.

Pregnant women between 16 and 22 weeks of gestation with an uncomplicated multiple pregnancy and an asymptomatic CL below 38 mm were eligible. A cutoff of 38 mm was chosen as it was associated with a treatment benefit of a pessary in a comparable Dutch population of women with a twin pregnancy at higher risk of preterm birth. Exclusion criteria were prior sPTB before 34 weeks, a cervical cerclage in the current pregnancy, maternal age less than 18 years, death of one or both of the foetuses in this pregnancy, major congenital abnormalities identified in the current pregnancy (defined as conditions of prenatal origin that are present at birth, potentially impacting an infant's health, development and/or survival), prior participation in the Quadruple P study, vaginal blood loss or contractions, CL below 2 mm and/or cervical dilatation of 3 cm and above. Gestational age (GA) was determined by first-trimester ultrasound according to the Dutch national guidelines [22].

## Randomisation and masking

CL was measured with a transvaginal ultrasound between 16 and 22 weeks during routine foetal scans, following the Society for Maternal and Fetal Medicine criteria [23]. All sonographers were trained according to national guidelines and used ultrasound systems meeting the Institute of Health and Environment's quality requirements [24,25]. Eligible women were counselled by trained nurses, midwives, or obstetric trainees or specialists. After providing written informed consent, pregnant individuals were randomised to receive either a cervical pessary or vaginal progesterone in a 1 to 1 ratio with a permuted block size of 2 and 4, stratified per centre to reduce the risk of confounding due to centre-specific practices. The unborn multiples were automatically assigned to the randomisation group of their mother. Randomisation was initially centrally controlled using ALEA, an online computerised randomisation service (ALEA Clinical software version 16, Forms Vision, Abcoude, the Netherlands) and from July 2022 Castor EDC (Electronic Data Capture v2022.3.2.0). Due to the nature of the interventions, participants, study staff or treating professionals were not blinded to allocation.

## Procedures

In participants allocated to the pessary group, an Arabin cervical pessary [26] (CE0482, MED/CERT ISO 9003/EN 46003; Dr Arabin GmbH and Company, KG; Witten, Germany) was placed in situ and removed at 36 weeks or in case of ruptured membranes or signs of infection or preterm labour, whatever came first. Insertion was done by experienced midwives or obstetricians, the majority of whom previously participated in similar trials using pessaries like the ProTwin trial and Quadruple P singletons trial [12,20]. The participating hospitals were provided with instructions on pessary placement. Three different pessary sizes were available (small: 65/25/32 mm; medium 70/25/32 mm; and large 70/25/35 mm). The required size was determined and confirmed through physical examination.

Participants allocated to the progesterone group self-administered 200 mg vaginal capsules (Utrogestan) daily until 36 weeks or earlier in case of ruptured membranes or PTB. They were informed by their obstetrician or research midwife how to insert the vaginal tablets, the preferred time (before sleeping) and the use of a medication diary.

For both groups, follow-up checks were done during antenatal visits scheduled within routine clinical care, with special attention to adherence, complaints, adverse symptoms or problems regarding usage, which was noted in the electronic patient file. If necessary, any issues with the pessary were checked by vaginal examination and managed by repositioning, removal, or replacement. Compliance to progesterone was monitored, and a vaginal examination was performed in participants with severe complaints.

Both groups received routine care according the local protocol without behavioural or physical restrictions. No standard physical or ultrasound follow-up examination was performed. Additionally, no double therapy was given, (i.e., no additional progesterone with a pessary or vice versa) and emergency cerclage decisions were made by the treated obstetrician.

Data on maternal characteristics, medical history, pregnancy, birth, and maternal and neonatal outcomes were collected in electronic case report forms (Open Clinica and Castor EDC). Participants and their offspring were followed until 10 weeks post-due date.

## Outcomes

The primary outcome was a composite adverse perinatal outcome containing periventricular leukomalacia (PVL)> grade 1, chronic lung disease (severe respiratory distress syndrome or bronchopulmonary dysplasia), intra ventricular haemorrhage grade III or IV, necrotising enterocolitis >stage 1, proven sepsis, stillbirth and death of the baby before discharge from the hospital (perinatal death is defined as death from GA of randomisation to 10 weeks after estimated due date). Outcomes were ascertained by qualified neonatologists. Secondary outcomes included time to delivery, rate of PTB at less than 24, 28, 32, 34 and 37 weeks (spontaneous, iatrogenic and total), premature rupture of the membranes, mode of delivery and placed cerclages. For the neonate, we compared birth weight (in grams), individual components of the

composite neonatal outcome, patent ductus arteriosus (PDA), neonatal seizures and admission days in neonatal intensive care unit. For the mother, we compared maternal morbidity (thrombo-embolic complications, infections (defined as genital tract infections, urinary tract infections, chorioamnionitis treated with antibiotics or suspected infections during labour for which antibiotics were given), pneumonia, endometritis and eclampsia or HELLP, vaginal blood loss, excessive vaginal discharge and maternal death. Among the secondary outcomes, several were not prespecified in the original protocol but were added to the statistical analysis plan due to their clinical importance, including (s)PTB before 24 weeks, premature rupture of membranes, mode of delivery, placement of cerclages, birth weight, individual components of the composite outcome, patent ductus arteriosus, neonatal seizures, twin-to-twin transfusion syndrome, vaginal blood loss, and excessive vaginal discharge. Regarding the secondary outcomes that were pre-specified in the protocol but not reported in the manuscript (namely, admission days for preterm labour and cost outcomes), we chose to address these in a separate cost-effectiveness analysis (CEA).

Serious adverse events were defined as maternal death, life-threatening events, events requiring hospitalisation (for complications that were not inherent to pregnancy), events resulting in persistent or substantial disability or incapacity, or any other serious or unexpected adverse event.

## Sample size

Based on available data at the time of protocol development, we expected 24% of adverse perinatal outcomes in the vaginal progesterone group on child level [27]. Based on the results of the Pro-Twin trial, we expected an adverse event rate of 12% in the pessary group [12]. Therefore, in this trial, we aimed for a reduction of 50% in the primary outcome, from 24% to 12%. Using a 5% type 1 error, a 20% type 2 error and a lost-to-follow-up rate of 5%. This resulted in a sample size of 332 pregnant individuals (166 per arm). Clustering of outcomes within multiple pairs was not accounted for.

## Statistical analysis

Analysis was done according to a prespecified statistical analysis plan (S3 File: Analysis Plan). The primary data analysis was according to the intention-to-treat principle. The primary outcome, the adverse perinatal outcome on child level, is presented in incidence rates with relative risks (RRs) and 95% confidence intervals (CIs) and $p$-values using a generalised estimating equations (GEE) with log link and binomial distribution with the robust estimator as covariance matrix and an independent working correlation matrix, estimating both crude rates and adjusted rates with centre as fixed covariate and mother as random effect to account for clustering of the multiples [28]. Adjustment of $p$-values was not performed, as no interim analysis for efficacy was performed. To ensure robustness of our primary analysis, a sensitivity analysis using a mixed-effects model including random effects for centre and mother was conducted. $P$-values <0.05 were supposed to indicate statistical significance.

In secondary analyses, if the difference in RR between crude and adjusted outcomes for the primary endpoint was less than 10%, centre was not included as a covariate. Statistical tests used two-sided $p$-values of 0.05 and 95% CIs.

At the neonatal level, outcomes were assessed using GEE to account for clustering within the mother, comparable to the primary outcome, with dichotomous outcomes analysed to calculate RRs, 95% CIs, and $p$-values and continuous outcomes were analysed using GEE with an identity link with mean differences and the corresponding 95% CIs along with the $p$-value reported [28].

Maternal outcomes were assessed for the total population of mothers using a generalised linear log-binomial model. Other dichotomous outcomes were assessed in the same way, while continuous outcomes were visually inspected and analysed based on their distribution (mean differences or medians with interquartile ranges). Time to delivery was evaluated by Kaplan–Meier estimates.

A per-protocol analyses was performed to evaluate robustness of the primary analysis including participants whose allocated treatment was continued up to 36 weeks of gestation or until (threatened) preterm delivery. Participants who

 

received a cerclage or switched treatment modality were not included in this per protocol analyses. An additional sensitivity analysis of the per protocol population assessed treatment intensity, using different cut-off points (60%, 70%, 80%, and 90%) for good adherence of proportion of days covered (PDC) as adherence measure [29–31].

Predefined subgroup analyses were performed on CL (25 mm and below versus above 25 mm), parity (multiparous women without previous sPTB between 23 and 37 weeks GA versus those with previous sPTB between 34 and 37 weeks GA and nulliparous women), chorionicity (monochorionic versus dichorionic pregnancies), number of foetuses (3 or more versus 2 foetuses), and GA (born at or before 36 + 0 weeks GA versus after 36 weeks GA). The rates of sPTB before 28 and 34 weeks were also assessed in the subgroup analyses, as was time to delivery, which was evaluated using Kaplan-Meier curves. Additionally, PTB rates were explored across various gestational weeks and CL ranges on maternal level.

### Patients and public involvement

During the design and conduct phase of this trial, the study proposal was reviewed by Care4Neo, the Dutch neonatology patient association affiliated with the European Foundation for the Care of Newborn Infants. They deemed the topic of extraordinary importance and strongly supported the study (S5 File: Declaration of Intent). Additionally, we agreed to participate in the PROMPT collaboration planning meta-analysis with individual participant data [32]. Therefore, the primary and secondary outcomes were selected to align with the Core Outcomes in Women's Health (CROWN) initiative, specifically the subset focussed on evaluating interventions to prevent preterm birth (www.crown-initiative.org) [33]. The CROWN initiative involves extensive patient participation, and the outcome measures were established in collaboration with them.

### Results

From 29th July 2014 to 8th September 2023, 276 participants provided written informed consent and were randomly assigned to receive either cervical pessary (n = 138) or vaginal progesterone (n = 138) (Fig 1). Based on complete data of 237 participants, the DSMC advised on 27th March 2023 to halt recruitment due to futility reasons, namely an expected apparent challenge in establishing superiority of pessary treatment. Although no formal futility stopping rules based on the statistical significance of the effect of the treatment were prespecified in the statistical analysis plan, the DSMC considered continued recruitment scientifically unjustified based on the observed data. Following an independent review by the METC of the Amsterdam UMC, it was concluded that the DSMC's advice should be followed, leading to early termination of the trial. Follow-up ended in March 2024.

Five participants were inappropriately randomised for various reasons, including having a CL measured after 22 weeks GA (n = 2) or in retrospect not meeting the inclusion criteria at the time of randomisation (n = 3). In addition, of the seven participants a written informed consent form was not properly stored after randomisation and insufficient information was provided in their electronic patient files to confirm whether they had followed the correct informed consent procedure. Two participants withdrew consent for data collection. Ultimately, 262 participants were included in the intention-to-treat analysis (Fig 1).

Baseline characteristics were comparable (Table 1). Participants were randomised at a median GA of 19 weeks and 5 days in the pessary group and 19 weeks and 6 days in the progesterone group, with a mean CL of 29.0 mm (SD 7.3.mm) versus 30.2 mm (SD 6.2 mm), respectively. Pregnancies were monochorionic in 26% of the participants in the pessary group and in 31% of the participants in the progesterone group. A total of seven triplets were included (pessary n = 3, progesterone n = 4).

The primary outcome occurred in 53 of 269 (19.7%) children in the pessary group compared to 36 of 262 (13.7%) children in the progesterone group (adjusted RR 1.42 95% CI [0.84, 2.41]; p = 0.19) (Table 2). The rates of (s)PTB < 28, 32, 34 and 37 weeks did not differ significantly between both groups (Table 3). The per-protocol analysis did not reveal any different insights, even in the sensitivity analyses on different cut-off values for compliance (Table 2, Tables A and B in S1

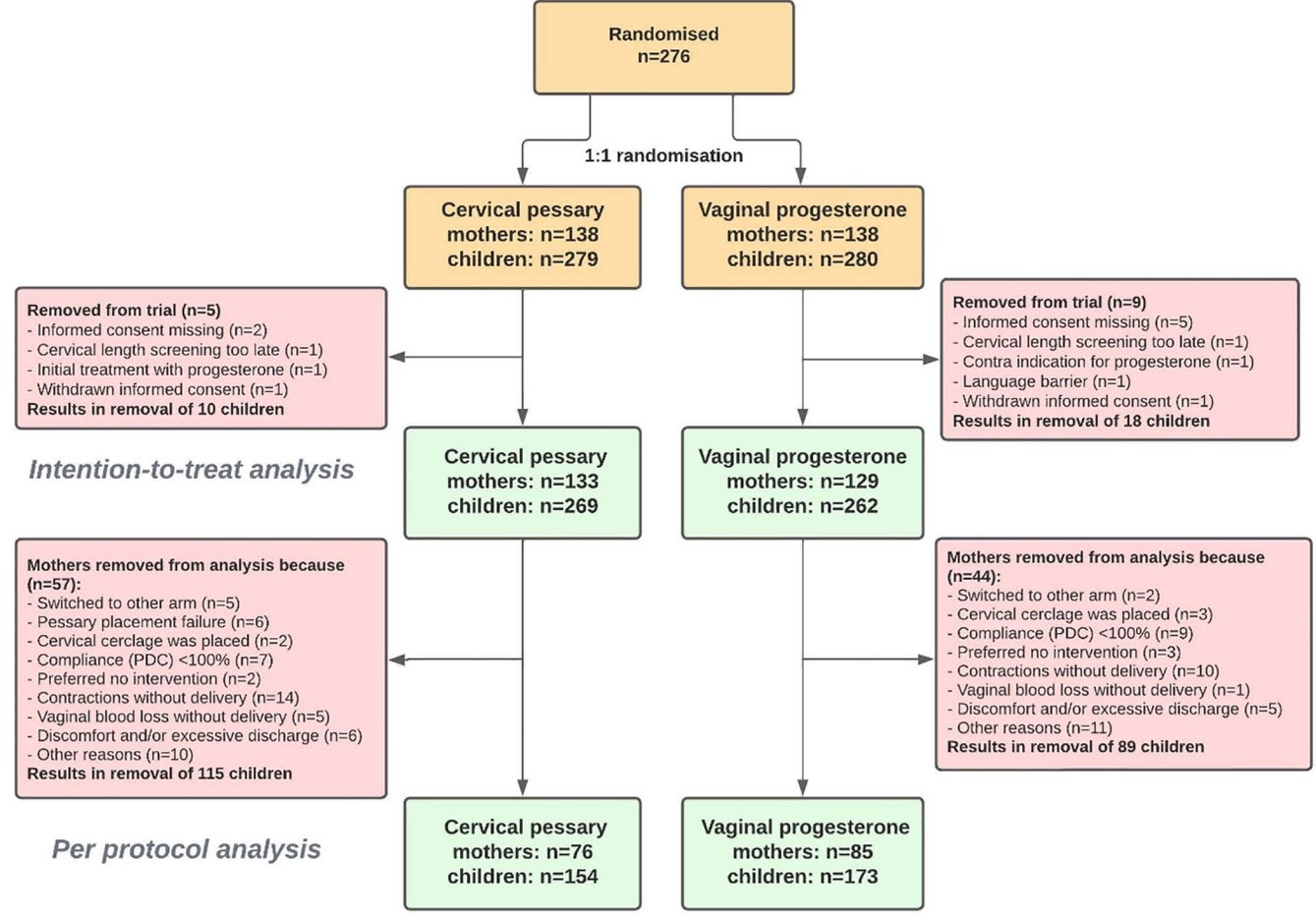

**Fig 1. Flowchart inclusions Quadruple P Multiples study.**

File). A sensitivity analysis on the variability across centres was comparable not statistically significant (*p* = 0.48), supporting the robustness and generalizability of the primary outcome.

Mean time to delivery was 96 days (SD 35) after pessary treatment and 99 days (SD 33) after progesterone (Table 3), and did not significantly differ as is shown by the Kaplan–Meier curve (Fig 2). Other neonatal outcomes were comparable between groups, including perinatal death (26/269 (9.7%) versus 21/262 (8.0%), RR 1.21, 95% CI [0.58, 2.51]; *p* = 0.62), duration of NICU admittance (18 days, IQR 6–38 versus 12 days, IQR 3–28, *p* = 0.25) and mean birth weight (2,054 gram versus 2,107 gram, *p* = 0.35) (Table 4).

At maternal level, significantly more often 'excessive discharge' was reported by participants in the pessary group compared to the progesterone group (31/133 (24.2%) versus 14/129 (11.2%), RR 2.16, 95% CI [1.21, 3.87]; *p* = 0.009) (Table 3). Vaginal bleeding was reported equally frequent in both groups (13/133 (9.8%) versus 13/129 (10.2%), RR 0.97, 95% CI [0.47, 2.01]; *p* = 0.93). Other maternal outcomes were also comparable between groups, including the incidence of genital tract infections (6/133 (4.6%) versus 6/129 (4.8%), RR 1.20, 95% CI [0.59,2.46]; *p* = 0.95) and chorioamnionitis (4/133 (3.1%) versus 4/129 (3.1%), RR 0.96, 95% CI [0.32,2.90]; *p* = 0.98). Emergency cervical cerclages were placed in two participants (1.5%) in the pessary group compared to 3 (2.3%) in the progesterone group (RR 0.71, 95% CI [0.14,3.75]; *p* = 0.69).

PLOS Medicine

**Table 1. Baseline characteristics.**

| Characteristic | Pessary N=133 | Progesterone N=129 |
|---|---|---|
| Maternal age, years (mean, SD) | 32.8 (5.0) | 31.34 (4.9) |
| Body-mass index, kg/m$^2$ (mean, SD) | 25.5 (5.8) | 24.4 (4.6) |
| Education Low* | 2 (4.3%) | 7 (13.0%) |
| Ethnicity | | |
| White | 79 (64.2%) | 81 (70.4%) |
| Black | 17 (13.8%) | 12 (10.4%) |
| Middle Eastern | 16 (13.0%) | 11 (9.6%) |
| Asian | 4 (3.5%) | 4 (3.5%) |
| Other | 7 (5.7%) | 7 (6.1%) |
| Current smoker | 4 (2.4%) | 7 (5.6%) |
| Uterus anomaly | 1 (0.8%) | 1 (0.8%) |
| Nulliparous | 74 (55.6%) | 77 (59.7%) |
| Previous preterm birth (34$^{+0}$–36$^{+6}$) | 3 (2.3%) | 2 (1.6%) |
| History of cervical surgery (Conisation/LLETZ) | 9 (7.0%) | 9 (7.0%) |
| History of curettage | 11 (9.2%) | 14 (11.8%) |
| Conception | | |
| Pregnancy after IVF/ ICSI | 13 (9.8%) | 14 (11.0%) |
| 3 or more foetus | 3 (2.3%) | 4 (3.1%) |
| Monochorionic pregnancy | 35 (26.3%) | 40 (31.0%) |
| GA (weeks + days) at randomisation (median, IQR) | 19+6 (18+6, 20+5) | 19+5 (19+0,20+5) |
| Cervical length at randomisation (mm) (mean, SD) | 28.97 (7.27) | 30.15 (6.15) |
| Cervical length range | | |
| 0–15 mm | 10 (7.5%) | 4 (3.1%) |
| 16–25 mm | 20 (15.0%) | 20 (15.5%) |
| 26–37 mm | 103 (77.4%) | 105 (81.4%) |
| Funnelling | 24 (18.0%) | 20 (15.5%) |

*Low education, Primary school, prevocational secondary education (VMBO in Dutch); LLETZ, large loop excision of the transformation zone; ISCI, intracytoplasmic sperm injections; IVF, in vitro fertilisation; GA, gestational age; IDR, Inter Quartal Range; SD, standard deviation.

Missing (N pes/ N prog): age (1/0), BMI (9/10), funnelling (11/11), education (87/74), ethnicity (10/14), curettage n= (14/9); conisation/LLETZ (4/2), uterus anomaly (4/6), smoking (6/5), conception (0/1).

**Table 2. Primary outcome: composite adverse neonatal outcome on child level.**

| | Pessary N=269 | Progesterone N=262 | RR [95% CI] | P-value |
|---|---|---|---|---|
| **Primary outcome on child level** | | | | |
| Composite adverse neonatal outcome (ITT), adjusted for mother | 53 (19.7%) | 36 (13.7%) | 1.43 [0.85, 2.43] | 0.18 |
| Composite adverse neonatal outcome (ITT), adjusted for centre and mother | 53 (19.7%) | 36 (13.7%) | 1.42 [0.84, 2.41] | 0.19 |

ITT, intention to treat; RR, relative risk; CI, confidence interval.

Outcomes were assessed using generalised estimating equations (GEE) with log link and binomial distribution with the robust estimator as covariance matrix and an independent working correlation matrix, estimating both crude rates and adjusted rates with centre as fixed covariate and mother as random effect to account for clustering of the multiples.

Table 3. Secondary outcomes on maternal level.

| | Pessary N=133 | Progesterone N=129 | RR [95% CI] | P-value |
|---|---|---|---|---|
| Composite adverse neonatal outcome (ITT), crude | 29 (21.8%) | 21 (16.3%) | 1.34 [0.81,2.22] | 0.26 |
| Composite adverse neonatal outcome (ITT), adjusted for centre | 29 (21.8%) | 21 (16.3%) | 1.30 [0.78,2.17] | 0.31 |
| **Obstetric outcomes** | | | | |
| Cerclage placement | 2 (1.5%) | 3 (2.3%) | 0.71 [0.14,3.75] | 0.69 |
| PROM | 47 (35.3%) | 47 (36.4%) | 0.97 [0.7,1.34] | 0.86 |
| Caesarean section | 125 (46.5%) | 102 (38.9%) | 1.19 [0.90,1.58] | 0.22 |
| Time to delivery (days), median (IQR)^ | | | | |
| PTB<37 weeks | 95 (73.6%) | 95 (71.4%) | 0.96 [0.82,1.12] | 0.60 |
| sPTB<37 weeks | 60 (46.5%) | 60 (45.1%) | 0.93 [0.71,1.21] | 0.60 |
| PTB<34 weeks | 47 (35.3%) | 38 (29.5%) | 1.16 [0.81,1.67] | 0.42 |
| sPTB<34 weeks | 37 (27.8%) | 35 (27.1%) | 0.99 [0.66,1.49] | 0.97 |
| PTB<32 weeks | 30 (22.6%) | 27 (20.9%) | 1.03 [0.65,1.66] | 0.89 |
| sPTB<32 weeks | 28 (21.1%) | 25 (19.4%) | 1.07 [0.66,1.74] | 0.78 |
| PTB<28 weeks | 20 (15.0%) | 15 (11.6%) | 1.30 [0.70,2.39] | 0.41 |
| sPTB<28 weeks | 20 (15.0%) | 14 (10.9%) | 1.38 [0.74,2.60] | 0.31 |
| PTB<24 weeks | 8 (6.0%) | 6 (4.7%) | 1.23 [0.44,3.46] | 0.70 |
| sPTB<24 weeks | 8 (6.0%) | 6 (4.7%) | 1.23 [0.44,3.46] | 0.70 |
| **Maternal outcomes** | | | | |
| Maternal mortality | 0 (0.0%) | 0 (0.0%) | NA | NA |
| Maternal morbidity | | | | |
| Thromboembolic complication | 0 (0.0%) | 0 (0.0%) | NA | NA |
| Pre-eclampsia/HELLP | 14 (10.5%) | 9 (7.0%) | 1.51 [0.68, 3.36] | 0.32 |
| Pneumonia | 0 (0.0%) | 0 (0.0%) | NA | NA |
| Chorioamnionitis* | 4 (3.1%) | 4 (3.1%) | 0.98 [0.25,3.85] | 0.98 |
| Urinary tract infections* | 15 (11.6%) | 12 (9.7%) | 1.20 [0.59, 2.46] | 0.62 |
| Genital tract infections* | 6 (4.6%) | 6 (4.8%) | 0.96 [0.32,2.90] | 0.95 |
| Endometritis or pelvic infection* | 5 (3.9%) | 4 (3.3%) | 1.19 [0.33,4.34] | 0.79 |
| Suspected infection during labour* | 17 (13.1%) | 8 (6.7%) | 1.96 [0.88,4.38] | 0.10 |
| Vaginal blood loss* | 13 (9.8%) | 13 (10.2%) | 0.97 [0.47,2.01] | 0.93 |
| **Excessive discharge*** | **31 (24.2%)** | **14 (11.2%)** | **2.16 [1.21,3.87]** | **0.009** |

ITT, intention to treat; RR, relative risk; CI, confidence interval; PROM, premature rupture of membranes; HELLP, haemolysis, elevated liver enzymes and low platelets.

Dichotomous outcomes were assessed using generalised linear log-binomial model.

^Calculated by Mann–Whitney U-test.

*Missing data (N): chorioamnionitis (6), urinary tract infection (7), genital tract infection (9), endometritis or pelvic infection (10), suspected infection during labour (12), vaginal blood loss (2), excessive discharge (9).

Among all randomised participants, serious adverse events occurred in 5 (3.8%) allocated to pessary and 1 (0.8%) in the progesterone group (RR 4.85, 95% CI [0.57,40.9]; p = 0.15) (Table D in S1 File). The most serious adverse events involved two maternal ICU admissions in the pessary group, one due to renal failure from pre-eclampsia/HELLP and one admission due to sepsis from severe chorioamnionitis. The other four hospital admissions were for non-obstetric indications (tachycardia, headache, severe bladder lesion after caesarean and urinary incontinence in known cauda equina syndrome). None of these serious adverse events was considered to be associated with the allocated treatment.

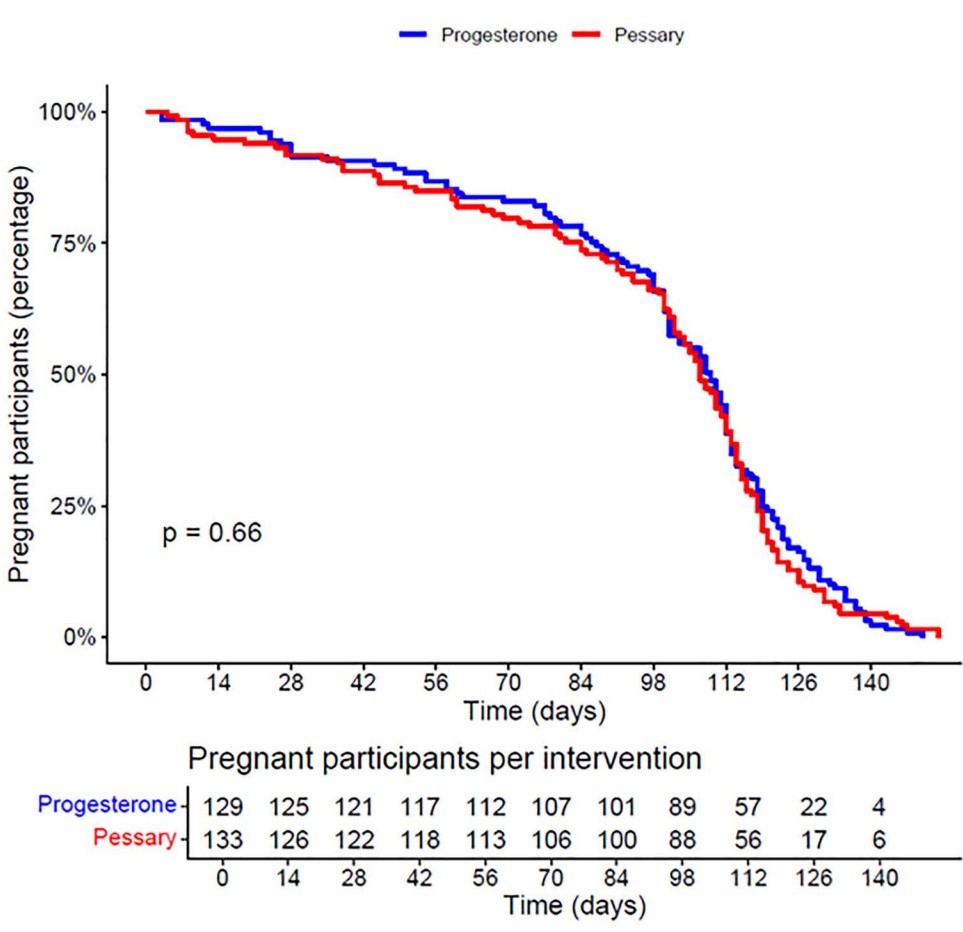

**Fig 2. Kaplan–Meier curve for time to delivery.**

In the predefined subgroup analyses, effect modification was seen for nulliparous participants compared to multiparous participants (with a history of a term birth or prior sPTB between 34 and 37 weeks' gestation). In nulliparous participants, the composite outcome occurred more often in the children in the pessary group (45/150) compared to the progesterone group (25/157) (30.0% versus 15.9%, RR 1.88 95% CI [1.03,3.43], interaction $p = 0.93$) (Table 5). In the other subgroup analyses (monochorionic versus dichorionic, CL below 25 mm versus above 25 mm, twins versus triplets and born before 36 weeks' gestation versus after 36 weeks), no effect modification was seen and thus no significant group differences could be identified (Tables 5 and 6). Table E in S1 File shows the exploratory analysis on PTB rates for CL 0–25 mm and above 25 mm, in which no notable differences were seen.

## Discussion

This RCT showed that in asymptomatic women with multiple pregnancies without a prior sPTB before 34 weeks' gestation and a shortened CL, the use of a cervical pessary did not result in improved perinatal outcomes compared to vaginal progesterone (19.7% versus 13.7%, RR 1.43; 95% CI [0.85,2.4]; $p = 0.18$). Among nulliparous participants, the observed differences showed a trend in favour of progesterone (30.0% versus 15.9%, RR 1.88; 95% CI [1.03,3.43], interaction

**Table 4. Secondary outcomes on neonatal level.**

| | Pessary N = 269 | Progester-one N = 262 | RR [95% CI] | P-value |
|---|---|---|---|---|
| Composite adverse neonatal outcome (PP), adjusted for mother and centre (100% PDC#) | 53 (19.5%) | 36 (13.3%) | 1.47 [0.74,2.91] | 0.28 |
| Neonatal outcomes | | | | |
| Birth weight (g), mean (SD)* | 2054 (777) | 2,107 (741) | 0.081 [−0.09,0.25] | 0.35 |
| Birthweight <2,500 g (%) | 175 (65.5%) | 168 (64.4%) | 1.02 [0.87,1.19] | 0.82 |
| Birthweight <1,500 g (%) | 62 (23.2%) | 50 (19.2%) | 1.21 [0.77,1.91] | 0.41 |
| Neonatal diagnosis | | | | |
| Patent ductus arteriosus | 9 (3.3%) | 8 (3.1%) | 1.10 [0.30,4.04] | 0.89 |
| Treated seizures | 1 (0.4%) | 0 (0%) | NA | NA |
| Chronic Lung disease # | 24 (8.9%) | 13 (5.0%) | 1.80 [0.73,4.43] | 0.20 |
| PVL > grade 1# | 0 (0%) | 0 (0%) | NA | NA |
| IVH grade III or IV# | 1 (0.4%) | 1 (0.4%) | 0.97 [0.062,15.41] | 1.00 |
| NEC > stage 1# | 5 (1.9%) | 7 (2.7%) | 0.70 [0.37,2.74] | 0.60 |
| ROP | 6 (2.2%) | 2 (0.8%) | 2.92 [0.34,25.15] | 0.61 |
| Culture-proven sepsis# | | | | |
| <72 hours after birth (early) | 1 (0.4%) | 0 (0.0%) | NA | NA |
| >72 hours after birth (late) | 15 (5.6%) | 9 (3.4%) | 1.62 [0.64,4.15] | 0.31 |
| Perinatal death# | 26 (9.7%) | 21 (8.0%) | 1.21 [0.58,2.51] | 0.62 |
| NICU admission (days), median (IQR) | 18 (6–38) | 12 (3–28) | −3.0 [−8.,1.5] | 0.25 |
| Congenital abnormalities | 6 (2.2%) | 2 (0.8%) | 2.94 [0.61,14.33] | 0.18 |

Severe respiratory distress syndrome (RDS) or Bronchopulmonary Dysplasia (BPD); PVL, periventricular leukomalacia; IVH, intraventricular haemorrhage; NEC, necrotising enterocolitis; ROP, retinopathy of prematurity; NICU, neonatal intensive care unit; PP, per protocol; PDC, proportion of days covered; RR, relative risk; CI, confidence interval.

#Part of composite outcome.

^Dichotomous outcomes were assessed using generalised estimating equations (GEE) to account for clustering within the mother. Continuous outcomes were analysed using GEE with an identity link, calculating mean differences with 95% CI.

*Missing data (N): birthweight: (3 children).

*p* = 0.93). Excessive discharge was more often reported in the cervical pessary group compared to the vaginal progesterone group (24.2% versus 11.2%, RR 2.16, 95% CI [1.21, 3.87], *p* = 0.009).

When we place the results in context, it is noticeable that the result of our trial showed the opposite effect of what we expected. We anticipated to find a reduction of the composite outcome to 12%, based on previous findings in the ProTwin trial [12]. In this study, in the subgroup with a CL below 38 mm, pessary treatment significantly reduced frequency of poor perinatal outcome and very preterm delivery compared to a control group. Considering our populations were assumed to be comparable and similar cut-off value for short CL were applied, we expected comparable outcomes. Instead, we found a higher incidence of composite adverse perinatal outcomes in the pessary group compared to the progesterone group (19.7% versus 13.7%). In addition, the composite adverse outcome in the pessary group in this trial was twice as high compared to the ProTwin pessary group (19.7% compared to 10.0%). Comparing study populations at baseline, 91% of the ProTwin population was white and 37% became pregnant after fertility treatment compared to 64% and 9.8%, respectively, in the QP trial. However, it is challenging to quantify the extent to which these baseline differences contributed to the differences in outcomes between the trials. This trial was subsequently halted due to concerns about futility, as it appeared highly unlikely that the superiority of the pessary could be demonstrated with the remaining number of participants. However, this decision leaves us uncertain whether the observed effect, specifically, better

**Table 5. Subgroup analyses of parity, chorionicity, number of foetuses and gestational age at birth on the primary outcome composite adverse perinatal outcome.**

| | Pessary | Progesterone | RR [95% CI] | P-value interaction term |
|---|---|---|---|---|
| **Composite adverse perinatal outcome (primary outcome)** | | | | |
| **Obstetric history** | | | | |
| Nulliparous women | 45/150 (30.0%) | 25/157 (15.9%) | 1.88 [1.03,3.43] | 0.93 |
| Multiparous sPTB 34$^{+0}$–36$^{+6}$ weeks | 2/6 (33.3%) | 2/4 (50.0%) | 0.67 [0.080,5.54] | |
| Multiparous no sPTB | 6/113 (5.3%) | 9/101 (8.9%) | 0.60 [0.18,1.96] | 0.09 |
| **Chorionicity** | | | | |
| Dichorionic | 43/199 (21.6%) | 24/178 (13.5%) | 1.60 [0.86,2.99] | 0.43 |
| Monochorionic | 10/70 (14.3%) | 12/84 (14.3%) | 1.00 [0.37,2.73] | |
| **Number of foetuses** | | | | |
| 2 foetuses | 49/260 (18.8%) | 36/250 (14.4%) | 1.31 [0.77,2.23] | NA |
| 3 or more foetuses | 4/9 (44.4%) | 0/12 (0.0%) | NA | |
| **Gestational age at birth** | | | | |
| Born < 36 weeks | 52/143 (36.4%) | 36/138 (26.1%) | 1.39 [0.86,2.25] | NA |
| Born ≥ 36 weeks | 1/126 (0.8%) | 0/124 (0.0%) | | |

RR, relative risk; CI, confidence interval; sPTB, spontaneous preterm birth.

**Table 6. Subgroup analyses on the primary outcome composite adverse perinatal outcome, sPTB < 34 and sPTB < 28 weeks.**

| | Pessary | Progesterone | RR [95% CI] | P-value interaction term |
|---|---|---|---|---|
| **Cervical length** | | | | |
| **Composite adverse perinatal outcome (primary outcome on child level)** | | | | |
| ≤25 mm | 25/61 (41.1%) | 17/49 (34.7%) | 1.18 [0.60,2.33] | 0.63 |
| >25 mm | 28/208 (13.5%) | 19/213 (8.9%) | 1.51 [0.73,3.12] | |
| **Spontaneous PTB < 34 weeks (secondary outcome)** | | | | |
| ≤25 mm | 37/61 (60.7%) | 26/49 (53.1%) | 1.14 [0.71,1.84] | 0.83 |
| >25 mm | 38/208 (18.3%) | 44/213 (20.7%) | 0.88 [0.51,1.54] | |
| **Spontaneous PTB < 28 weeks (secondary outcome)** | | | | |
| ≤25 mm | 19/61 (31.1%) | 16/49 (32.7%) | 0.95 [0.43,2.10] | 0.39 |
| >25 mm | 22/208 (10.6%) | 12/213 (5.6%) | 1.88 [0.72,4.89] | |

RR, relative risk; CI, confidence interval; PTB, preterm birth.

outcomes with progesterone treatment, might have reached statistical significance. If so, the findings could have been even more compelling.

This study is one of the few multicentre RCTs that directly compares the effectiveness of cervical pessary to vaginal progesterone in preventing PTB in women with a multiple pregnancy and an asymptomatic cervix of below 38 mm. Since these women had no history sPTB before 34 weeks' gestation, they had not been offered preventive progesterone treatment during pregnancy. A cut-off of CL at 38 mm allowed for assessment of effectiveness of these treatments across a broader range of CLs, thereby assessing a larger at-risk population than would have been possible with a 25 mm cut-off. This approach makes the findings more applicable to a general pregnant population. Another important finding is that the percentages of sPTB was very high, particularly among participants with a CL of 25 mm or less. Independent of the

intervention, 30% of this group delivered spontaneously before 28 weeks' gestation, which has major implications for neonatal outcomes. This underscores the importance of measuring CL in the second trimester, also for this group of pregnant women.

The study also has limitations. First of all, the nature of the interventions made blinding impossible, potentially introducing bias. Additionally, self-reported medication compliance in the progesterone group was low, with fewer than 30% of participants returning their medication diaries. As a result, obstetric caregivers' notes and verbal reports to research nurses were used to assess adherence, which may have led to an overestimation of actual compliance and an underestimation of progesterone's preventive potential in the per-protocol analysis. Given that our findings already indicate that pessary treatment is not superior to progesterone, we can therefore assure the robustness of this trial's conclusion. Furthermore, per-protocol analyses are inherently prone to confounding, as women who adhered may differ systematically from those who did not, and the findings should therefore be interpreted with caution. Another limitation concerns its design. While at the time of protocol development and in previous similar studies the trial was not considered as a cluster randomised design, we have corrected for clustering in the current analyses, which we consider sufficient; nevertheless, we acknowledge that in future studies such clustering should be addressed from the design phase, and that the lack of initial adjustment may have led to some degree of underpowering. Finally, the accuracy of the cut-off values for determining good or poor adherence remains challenging, given the uncertainty regarding the minimum usage requirement necessary to achieve a positive outcome with the interventions used [29–31]. We observed that the incidence of composite adverse perinatal outcomes did not differ between those with 100% adherence to those with 60% adherence in both intervention groups. This raises the question of whether 60% adherence is already sufficient for protection, or if the intervention itself may have limited effectiveness.

One other RCT directly compared a cervical pessary to vaginal progesterone in multiple pregnancies, with methods comparable to ours [14]. They found a significant reduction in composite adverse perinatal outcomes in the pessary group compared to the progesterone group (19% versus 27%; RR 0.70, 95% CI [0.43, 0.93]). However, this was a secondary outcome and did not include PVL. Our results show a similar outcome rate in the pessary group (19.7%), but a lower rate in the progesterone group (13.7%). Despite the benefit of the pessary on the composite outcome, Dang and colleagues found no significant benefit on PTB before 34 weeks (pessary 16% versus progesterone 22%; RR 0.73, 95% CI [0.46, 1.18]), similar to our findings. The main difference between these two trials is the progesterone dosage, 200 mg in our trial versus 400 mg in the trial of Dang and colleagues. The optimal progesterone dosage for twin pregnancies remains uncertain. Additionally, the lowest measured CL in Dang's study was 18 mm, with funnelling occurring only in 5% of the pessary group. In our trial, 7.5% had a CL of 15 mm or below, with 18% funnelling in the pessary group, suggesting a higher baseline risk for sPTB in addition to a reduced pessary effectiveness in shorter cervixes. However, higher rates of adverse events were also seen in the progesterone group with a shorter cervix. Lastly, population differences, such as higher percentage of nulliparous participants (86.5%), higher education levels and conception via IVF/ICSI (94.0%) in the study from Dang and colleagues, may contribute to the variation in outcomes, though the extent remains unclear.

Since no other trials directly compare cervical pessary and vaginal progesterone in multiple pregnancies, we can only draw comparisons with singleton populations or meta-analyses comparing these interventions against expectant management in multiple pregnancies. In the Quadruple P singletons trial, similar findings were reported, with no superiority of a cervical pessary over vaginal progesterone in low-risk participants without a history of sPTB before 34 weeks and an asymptomatic short cervix [20]. However, progesterone showed an advantage over pessary in the subgroup of a short cervix of 25 mm or less. This effect was not seen in the multiples.

The most recent meta-analysis comparing a pessary to expectant management of Norman and colleagues including five randomised trials concluded that a cervical pessary did not significantly reduce PTB before 34 weeks' gestation in patients with twin pregnancies and a short cervix regardless of their obstetric history and no differences were found in

neonatal morbidity and mortality [17]. On the other hand, a recent meta-analysis by Conde-Agudelo and colleagues of six randomised trials comparing vaginal progesterone with expected management found that vaginal progesterone significantly reduced the risk of PTB before 28–34 weeks and neonatal morbidity and mortality in twin pregnancies with a CL of 30 mm or less and 25 mm or less compared to standard care [10]. Additionally, two large trials were recently halted due to significantly higher rates of perinatal mortality in the pessary groups compared to cervical cerclage, although these were in high-risk singleton populations [34,35]. Given these negative perinatal outcomes, the lack of evidence of superiority of the pessary to progesterone in our trial, along with the absence of a significant effect of the pessary in the meta-analyses, there is currently no evidence to support the use of a cervical pessary for preventing PTB and adverse perinatal outcomes in multiple pregnancies.

Our research findings provide an opportunity to update an individual patient data meta-analysis on the efficacy of both pessary and progesterone use in preventing PTB. Since we cannot demonstrate the superiority of the pessary, it is essential to explore which methods are more effective in managing these high-risk pregnancies. While progesterone may have a positive effect, although it may not be large. Currently, the attention in prevention of PTB in multiple pregnancies seems to be shifting to a third intervention: cerclage, both ultrasound-indicated and physical examination-indicated. Despite initial concerns about its safety in multiple pregnancies, recent studies have shown positive outcomes [36,37]. Consequently, the TWIN-Cerclage study [38] has been initiated to investigate the effect of ultrasound-guided and physical examination-indicated cerclage in twins with a short cervix of 25 mm or less on PTB and adverse perinatal outcomes.

To summarise, our study did not find superiority of pessary treatment over progesterone in the prevention of a composite adverse perinatal outcome in multiple pregnancies without a prior sPTB at less than 34 weeks' gestation and with a midtrimester short cervix below 38 mm. In the subgroup analysis of nulliparous women, a pessary seemed less effective than progesterone in preventing a composite neonatal outcome.

## Supporting information

**S1 File. Supporting tables Quadruple P multiples study. Table A:** Protocol amendments and deviations. **Table B:** Details of intervention use: reasons for premature termination of allocated intervention on maternal level. **Table C**: Per protocol analyses for various compliance thresholds of proportion of days covered (PDC) on child level. **Table D:** Serious Adverse Events. **Table E**: Exploratory analysis for cervical length ≤25 mm and >25 mm on the secondary outcomes (s) PTB < 37, (s)PTB < 34, (s)PTB < 32, (s)PTB < 28 and (s)PTB < 24 weeks.
(PDF)

**S2 File. Study protocol.**
(PDF)

**S3 File. Statistical analysis plan.**
(PDF)

**S4 File. Study approval.**
(PDF)

**S5 File. Declaration of intent.**
(PDF)

**S1 CONSORT Checklist.** This checklist is licensed under the Creative Commons Attribution 4.0 International License (CC BY 4.0; https://creativecommons.org/licenses/by/4.0/).
(PDF)

## Acknowledgments

We would like to acknowledge and thank all women who participated in this study, as well as the staff of the Dutch trial bureau of the Dutch Consortium, all participating hospitals and especially all the research nurses involved in this trial, especially J.J.H. Bakker and L. Videler-Sinke. We would also like to acknowledge and thank the other members of the Quadruple P Research Group, namely the local principal investigators from the hospitals that were initially involved, but withdrew their participation in the trial and stopped recruiting new patients: M.N. Bekker, University of Utrecht, Wilhelmina's Children Hospital, UMC Utrecht, the Netherlands; J. Langeveld, Atrium Medisch Centrum Zuyderland, the Netherlands; D.N.N. Papatsonis, Amphia Ziekenhuis, the Netherlands; T.E.M. Verhagen, Slingeland Ziekenhuis, the Netherlands; E.C.J. Verheijen, Maasziekenhuis Pantein, the Netherlands; H. Visser, TerGooi Medisch Centrum, the Netherlands; Y.M. de Mooij, Zaans Medisch Centrum, the Netherlands en I.M. Evers, Meander Medisch Centrum, the Netherlands.

## Author contributions

**Conceptualisation:** Bouchra Koullali, Ben W. J. Mol, Brenda M. Kazemier, Eva Pajkrt.

**Formal analysis:** Charlotte E. van Dijk, Marijke C. van der Weide.

**Funding acquisition:** Eva Pajkrt.

**Investigation:** Charlotte E. van Dijk, Maud D. van Zijl, Bouchra Koullali, Eline S. van den Akker, Brenda J. Hermsen, Joris van Drongelen, Marjon A. de Boer, Wilhelmina M. van Baal, Karlijn C. Vollebregt, Flip W. van der Made, Sanne J. Gordijn, Marieke Sueters, Lia D. E. Wijnberger, Martijn A. Oudijk, Brenda M. Kazemier, Eva Pajkrt.

**Methodology:** Charlotte E. van Dijk, Bouchra Koullali, Marijke C. van der Weide, Ben W. J. Mol, Brenda M. Kazemier, Eva Pajkrt.

**Project administration:** Charlotte E. van Dijk, Annabelle L. van Gils, Maud D. van Zijl.

**Resources:** Eva Pajkrt.

**Software:** Charlotte E. van Dijk.

**Supervision:** Ben W. J. Mol, Eva Pajkrt.

**Validation:** Charlotte E. van Dijk, Annabelle L. van Gils, Maud D. van Zijl, Marijke C. van der Weide.

**Visualisation:** Charlotte E. van Dijk.

**Writing – original draft:** Charlotte E. van Dijk, Annabelle L. van Gils, Ben W. J. Mol, Brenda M. Kazemier, Eva Pajkrt.

**Writing – review & editing:** Charlotte E. van Dijk, Annabelle L. van Gils, Maud D. van Zijl, Bouchra Koullali, Marijke C. van der Weide, Eline S. van den Akker, Brenda J. Hermsen, Joris van Drongelen, Marjon A. de Boer, Wilhelmina M. van Baal, Karlijn C. Vollebregt, Flip W. van der Made, Sanne J. Gordijn, Marieke Sueters, Lia D. E. Wijnberger, Martijn A. Oudijk, Ben W. J. Mol, Brenda M. Kazemier, Eva Pajkrt.

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
