## [Editor Report · Decision Letter 0]

5 Mar 2025

Dear Dr van Dijk,

Thank you for submitting your manuscript entitled "Cervical pessary versus vaginal progesterone in women with a multiple pregnancy and a short cervix: a randomised controlled trial." for consideration by PLOS Medicine.

Your manuscript has now been evaluated by the PLOS Medicine editorial staff and I am writing to let you know that we would like to send your submission out for external assessment.

However, we first need you to complete your submission by providing the metadata that are required for full assessment. To this end, please login to Editorial Manager where you will find the paper in the 'Submissions Needing Revisions' folder on your homepage. Please click 'Revise Submission' from the Action Links and complete all additional questions in the submission questionnaire.

Please re-submit your manuscript within two working days, i.e. by Mar 07 2025 11:59PM.

Once your full submission is complete, your paper will undergo a series of checks in preparation for full assessment.

Kind regards,

Richard Turner, PhD

Consulting Editor, PLOS Medicine

plosmedicine@plos.org

---

## [Decision Letter · Decision Letter 1]

20 Jun 2025

Dear Dr van Dijk,

Many thanks for submitting your manuscript "Cervical pessary versus vaginal progesterone in women with a multiple pregnancy and a short cervix: a randomised controlled trial." (PMEDICINE-D-25-00762R1) to PLOS Medicine. The paper has been reviewed by subject experts and a statistician; their comments are included below and can also be accessed here: [LINK]

After discussing the paper with the editorial team and an academic editor with relevant expertise, I'm pleased to invite you to revise the paper in response to the reviewers' comments. We plan to send the revised paper to some or all of the original reviewers, and we cannot provide any guarantees at this stage regarding publication.

We ask that you submit your revision by Jul 10 2025 11:59PM. However, if this deadline is not feasible, please contact me by email, and we can discuss a suitable alternative.

Don't hesitate to contact me directly with any questions (lgaynor@plos.org).

Best regards,

Louise

Louise Gaynor-Brook, MBBS PhD

Senior Editor

PLOS Medicine

lgaynor@plos.org

Comments from the reviewers:

Reviewer #1: This thoughtfully conducted trial randomized multiple gestations to either pessary or progesterone and was ultimately stopped for futility. This trial is important in the overall literature and answers a key question of the role of pessary vs. progesterone in this important high-risk population. The overall methodology is sound and thought complex, the statistical analysis approach to addressing neonatal and maternal outcomes is robust. In particular, using an important outcome like morbidity instead of just surrogates like gestational age is appreciated. Similarly, the low-utilization of cerclage is a strength of this study.

The following suggestions are intended to further strengthen the study and are minor in nature:

1. Line 131- "main" would be better worded "primary"

2. Line 142- "case" should be "cases"

3. Line 161-Individuals do not need to be listed.

4. Line 196-A start date of the second randomization program but not the first.

5. Line 205 - If possible numerically what the words "most of whom" means.

6. Line 280- "to the other" either delete the word the or specify what the other means.

7. Line 425- Consider changing malacia to leukomalacia.

8. Line 604- The groups need to be labeled at the top.

Reviewer #2: This manuscript presents the results of a randomised trial where women pregnant with twins or triplets and an asymptomatic shortened cervix were randomised to receive a cervical pessary or vaginal progesterone. The primary outcome was adverse perinatal outcomes. My comments focus on the statistical aspects of this manuscript.

1. Given that mothers were randomised but the primary outcome was measured on the infant level, this is a cluster randomised trial. Please ensure that reporting aligns with the CONSORT extension for cluster randomised trials.

2. It is stated that this trial was stopped for futility, but there doesn't seem to be any allowance for that in the interim analysis section of the SAP. It is stated in the SAP that "there were no stopping rules based on the statistical significance of the effect of the treatment". How then was the trial stopped for futility? Please provide further details.

3. Was the sample size calculation based on the proportion of mothers who would experience adverse outcomes, or the number of babies who would experience adverse outcomes? If 332 participants is the number of mothers, then please clarify this. If the calculation was based on the number of babies, then the clustering of babies born to the same mother (i.e. multiple births) should have been accounted for.

4. Please provide more details regarding the randomisation. Were permuted blocks used? Was randomisation stratified by any variables?

5. I was glad to see that baby-level outcomes were analysed by fitting a marginal model via generalised estimating equations to account for clustering. However, I think how this approach is described can be clarified in the Statistical analysis section. For example "Child-level outcomes were analysed via a marginal model fitted using generalised estimating equations, with an exchangeable working correlation matrix to account for clustering of multiples within mothers". Please note that the assumed working correlation matrix structure needs to be specified - was an exchangeable working correlation matrix assumed?

6. It is stated that "a sensitivity analysis with a mixed model using a random effect for the centre was conducted, allowing the extrapolation of outcomes to other centres". However, a mixed model with random effects for centre would estimate a centre-specific effect, so it is unclear how this sensitivity analysis achieved the aim of allowing extrapolation to other centres. Please clarify - were the random effects marginalised over? How was clustering by mother accounted for in this analysis?

7. The description of the analysis on line 268 ("At the neonatal level…") does not seem to describe any different analysis to that described above. What is the difference? (Please note that the working correlation matrix and link function also needs to be specified for this analysis).

8. It is stated that continuous outcomes were analysed via t-tests etc. However - the clustering of outcomes within mothers would need to be accounted for in this analysis too, if the outcomes are measured on the infant level. Marginal models fit via GEE or conditional models with random effects for mother should be fit to analyse these outcomes.

9. A per-protocol analysis is described - however, such an analysis compares groups based on post-randomisation variables and thus is subject to confounding and not an appropriate method for comparing groups. More appropriate methods of analysis would be to estimate complier average causal effects. For more information and more appropriate analysis techniques to be applied please see https://doi.org/10.1016/j.jphys.2021.02.002 (for example - there are plenty of other papers on this). The switching of treatments is an intercurrent event, and requires careful consideration and analysis: please see the ICH E9 statistical principles for clinical trials scientific guideline and the primer available at https://doi.org/10.1136/bmj-2023-076316.

10. A few participants were inappropriately randomised - and it seems that these patients were all excluded from analyses. Whether this is the most appropriate course of action is questionable - as described in PMID: 26033877, application of the intention-to-treat principle would mean that these patients are included in analyses. Please provide results of analyses including data from these patients as a sensitivity analysis.

11. As stated in the CONSORT statement, hypothesis tests for baseline characteristics should not be conducted. Please remove the mention of the p-value in Table 1.

12. Please include the number of mothers in each group at the top of Table 1 and the names of the groups.

Reviewer #3: This is an important and well conducted RCT looking at the use of Arabian pessary vs progesterone for multiple pregnancies at risk of preterm birth.

The minor comments below will help to improve the manuscript:

Abstract

1. Line 61 - not necessary to include limitations sentence in abstract

Introduction

2. Line 132 - WHO definition of extremely preterm birth is <28 weeks, very preterm is <32 weeks, please modify this sentence to ensure correct definitions.

3. Line 143 - Ref 18 (EPPPIC) did not support progesterone use in multiples (RR 1.01). Please comment.

Methods

4. Line 181 - why do you use 38mm as a cut off, was it to be consistent with ProTwin Trial? I am interested to know how many multiples were not eligible for the study as in my experience, the vast majority will have cervical lengths below 38mm and not be at risk for preterm birth. My concern is that by including and treating the vast majority of multiples you are muddying the benefit of these therapies for those multiples actually at risk.

5. Line 182 - you have excluded women with a cerclage, how common is this practice in your country and what are the indications for cerclage in multiples in your units? Does this mean that the most at risk women (ie. this with CL <15mm) will have a cerclage and therefore, be excluded from this study?

6. Why did you chose to stratify based on centre?

7. Line 285 - I thought the exclusion criteria was no previous PTB <34 weeks? Please clarify your pre-defined sub group analysis and remove multiparous with prior delivery <34 weeks.

Discussion

8. Line 370 - can you comment on why the nulliparous women may show benefit of progesterone (cf pessary), compared with multiparous women?

9. Table 1 - no pessary and progesterone heading row, please add.

Reviewer #4: The authors report a RCT of pessary vs progesterone in multifetal gestation and a short cervix. The trial is well designed and performed; the reporting requirements are complete. The authors are to be congratulated on their work.

The trial was designed to provide clarity on how to reduce preterm birth and adverse outcomes in multifetal gestation. The literature is mixed, and the hope was to be able to perform a definitive trial.

Unfortunately, the trial results are confusing and instead add to the mix of contradictory findings. This is simply what they found, but it does not provide clarity for the clinician for this difficult topic.

The primary outcome is a composite of adverse perinatal outcomes. On page 9 these are listed as PVL>1, severe RDS/BPD, IVH ¾, NEC, sepsis, stillbirth, death before discharge. On page 23 the composite results are listed and are adjusted - but it is not clear what is actually in the composite and which are the primary drivers.

Is perinatal death both stillbirth and death before discharge or is it the typical definition of death between 22wks and 7 days of life?

Can the authors include in the table the composite and then below the composite the components of the composite?

Is death of one twin considered differently than death of both twins (or triplets)

Why was <38mm considered a short cervix in this population?

Why was prior sPTB <34 weeks an exclusion?

The anticipated adverse outcome was anticipated to be 12% in pessary, and it was ~19% in the trial, and the anticipated adverse outcome was anticipated to be 24% in progesterone, and it was ~13%. This is supremely confusing. Why? The authors attempt to address this in lines 374-390 but it is unfulfilling as they conclude if they had not stopped for futility perhaps, they would be able to provide more clarity.

Please provide more information on why the decision to halt was made with only 56 patients left to recruit - without the planned recruitment it leaves so many questions unanswered.

The conclusion is the authors did not find superiority of pessary for management over progesterone, indeed the pessary performed quite poorly in this trial.

How many discontinued the pessary?

How many discontinued the vaginal progesterone?

---

* Please upload any figures associated with your paper as individual TIF or EPS files with 300dpi resolution at resubmission; please read our figure guidelines for more information on our requirements: http://journals.plos.org/plosmedicine/s/figures. While revising your submission, please upload your figure files to the PACE digital diagnostic tool, https://pacev2.apexcovantage.com/. PACE helps ensure that figures meet PLOS requirements. To use PACE, you must first register as a user. Then, login and navigate to the UPLOAD tab, where you will find detailed instructions on how to use the tool. If you encounter any issues or have any questions when using PACE, please email us at PLOSMedicine@plos.org.

* Competing interests statement: Authors have declared no financial relationships with any organizations that might have an interest in the submitted work in the previous three years. Please note that everyone involved in the peer review process, including authors, editors and reviewers, must declare all potentially competing interests that occurred within 5 years of conducting the research under consideration, or preparing the article for publication. Interests outside the 5-year time frame must also be declared if they could reasonably be perceived as competing according to the definition above. Please revise your competing interests statement accordingly.

* Please note that data availability should not be time restricted ("Proposals for data access may be submitted up to 36 months following article publication"), restricted to specific future applications ("Data will be made available exclusively for individual participant data meta-analyses"), or have a study author as the contact person for data access requests. Directing data requests to a non-author institutional point of contact, such as a data access or ethics committee, helps guarantee long term stability and availability of data.

PLOS Medicine requires that the de-identified data underlying the specific results in a published article be made available, without restrictions on access, in a public repository or as Supporting Information at the time of article publication, provided it is legal and ethical to do so. The Data Availability Statement (DAS) requires revision.

For each data source used in your study:

FIGURES AND TABLES

SUPPLEMENTARY MATERIAL

REFERENCES

STUDY TYPE-SPECIFIC REQUESTS - RCTs

* PLOS Medicine requires that all trials be prospectively registered in one of registries recognized by WHO. Please ensure that study registration details are included in the Methods section.

* Please structure the Methods section using the following sub-headings: Study design and participants, Randomization and masking, Procedures, Outcomes, Statistical analysis.

* It is not made sufficiently clear in the main text that the trial did not reach its planned sample size i.e. that it was underpowered. Please comment on this detail in the main manuscript text.

* Inclusion/exclusion criteria: Death of one or both of the foetuses in this pregnancy not explicitly specified in the main manuscript text.

* The study protocol should include details of any amendments, as well as the date of their approval by the institutional review or ethics committee. Please also detail any deviations from the study protocol in the Methods section of your manuscript.

* Likewise, please confirm that the statistical analysis plan provided (version 2.0, dated 5th June 2024) is the final version of this document prior to data lock. Please note that version number and date of the statistical analysis plan differ on page 1 of the pdf; please clarify.

* From protocol version 3.0 (dated 17th April 2020) and the statistical analysis plan (version 2.0, dated 5th June 2024) we note the following:

- Secondary outcomes reported but not prespecified in the protocol:

PTB before 24 weeks

Premature rupture of the membranes

Mode of delivery

Placed cerclages

Birth weight

Individual components of the composite neonatal outcome

Patent ductus arteriosus (PDA)

Neonatal seizures

Vaginal blood loss

Excessive vaginal discharge

Twin-to-twin Transfusion Syndrome

- Secondary outcomes pre-specified in the protocol but not reported:

Admission days for preterm labour

Costs

* Please clarify and explain all discrepancies between the paper and protocol. If the outcomes were not prespecified in the protocol, please define them in the Methods (Outcomes section) as post hoc and explain why they were added. Post-hoc comparisons should be presented as hypothesis generating rather than conclusive.

* Please ensure that all prespecified outcomes (primary, secondary, and exploratory) are listed in the Methods/Outcomes section and indicate whether there are outcomes that are not presented in the current report.

* Please specify the dates (Month Day, Year) during which study enrollment and follow up occurred.

* Please include absolute numbers wherever you report percentages; eg, n/N (%)

* Please present the safety data for the study including numbers of specific events and whether or not adverse events are thought to be related to treatment. AEs should be reported in the abstract, per CONSORT and CONSORT-Harms.

* Please complete the CONSORT checklist (https://www.equator-network.org/reporting-guidelines/consort/) and ensure that all components of CONSORT are present in the manuscript, including how randomization was performed, allocation concealment, blinding of intervention, definition of lost to follow-up, power statement. When completing the checklist, please use section and paragraph numbers, rather than page numbers.

* Please report your abstract according to CONSORT for abstracts, following the PLOS Medicine abstract structure (Background, Methods and Findings, Conclusions) https://www.equator-network.org/reporting-guidelines/consort-abstracts/

* If your trial had to undergo important modifications in response to extenuating circumstances, please complete the CONSERVE-CONSORT checklist and provide in your Supporting Information; (https://www.equator-network.org/reporting-guidelines/guidelines-for-reporting-trial-protocols-and-completed-trials-modified-due-to-the-covid-19-pandemic-and-other-extenuating-circumstances-the-conserve-2021-statement/). When completing the checklist, please use section and paragraph numbers, rather than page numbers.

* In keeping with our commitment to Open Science, please include the study protocol document and analysis plan (including any amendments) as Supporting Information to be published with the manuscript if accepted.

* Please note that PLOS Medicine requires prospective, public registration of a data sharing plan (as part of mandatory clinical trials registration) for all clinical trials that began enrollment on or after January 1, 2019, in accordance with ICMJE requirements.

---

## [Decision Letter · Decision Letter 2]

19 Sep 2025

Dear Dr. van Dijk,

Thank you very much for re-submitting your manuscript "Cervical pessary versus vaginal progesterone in women with a multiple pregnancy and a short cervix: a randomised controlled trial." (PMEDICINE-D-25-00762R2) for review by PLOS Medicine.

I have discussed the paper with my colleagues and the academic editor and it was also seen again by 2 reviewers. I am pleased to say that provided the remaining editorial and production issues are dealt with we are planning to accept the paper for publication in the journal.

Please note that reviewer #2 has remaining requests. We discussed the issue of this being a clustered RCT with the academic editor, who stated "technically the trial is clustered as the infants are in the outcomes, but practically it would be more confusing to call it a cluster trial". Furthermore, we appreciate the 2nd comment from Reviewer #2, but we think the prespecified analysis needs to be reported on.

[LINK]

We look forward to receiving the revised manuscript by Sep 26 2025 11:59PM.   

Sincerely,

Suzanne De Bruijn, PhD

Associate Editor 

PLOS Medicine

plosmedicine.org

Requests from Editors:

GENERAL EDITORIAL REQUESTS

* Please confirm that your abstract complies with our requirements, including format (three sections: Background, Methods and Findings, and Conclusions) and providing all the information relevant to this study type https://journals.plos.org/plosmedicine/s/submission-guidelines#loc-abstract

* Please ensure that all abbreviations are defined at first use throughout the text.

* Please confirm that all numbers presented in the abstract are present and identical to numbers presented in the main manuscript text.

* We appreciate the inclusion of the files for the study approval, as well as supplement F "VOC declaration of intent regarding incubator infants.pdf" ; However, could you change the files in our system from 'supplement' to 'supporting', as otherwise they will get published with the paper.

GENERAL

* Please review your text for claims of novelty or primacy (e.g. 'for the first time') and remove this language. In addition, please check that any use of statistical terms (such as trend or significant) are supported by the data, and if not please remove them.

* Please remove the 'conclusions' subheading from the discussion. Please also remove any other subheadings from the discussion.

* Statistical reporting: Please revise throughout the manuscript, including tables and figures.

- Please report statistical information as follows to improve clarity for the reader " "22% (95% CI [13,28]; p</=)" ".

- Please separate upper and lower bounds with commas instead of hyphens as the latter can be confused with reporting of negative values.

- Please repeat statistical definitions (HR, CI etc.) for each set of parentheses.

-Author summary: Please include that the trial was halted for futility.

FUNDING STATEMENT

* The funding statement should include: specific grant numbers, initials of authors who received each award, URLs to sponsors’ websites. Also, please state whether any sponsors or funders (other than the named authors) played any role in study design, data collection and analysis, the decision to publish, or preparation of the manuscript. If they had no role in the research, include this sentence: “The funders had no role in study design, data collection and analysis, decision to publish, or preparation of the manuscript.”

* It appears that one or more study authors is affiliated with one or more of the agencies that funded the study. Thus, the statement “The funders had no role in study design, data collection and analysis, decision to publish, or preparation of the manuscript” does not apply. Please revise the Financial Disclosure accordingly, as in "[Author name] is [author's role] at [funding agency]. The funders had no other role in study design…..”

COMPETING INTERESTS STATEMENT

* All authors must declare their relevant competing interests per the PLOS policy, which can be seen here: https://journals.plos.org/plosmedicine/s/competing-interests For authors with ties to industry, please indicate whether any of the interests has a financial stake in the results of the current study.

DATA AVAILABILITY

* PLOS Medicine requires that the de-identified data underlying the specific results in a published article be made available, without restrictions on access, in a public repository or as Supporting Information at the time of article publication, provided it is legal and ethical to do so. Please see the policy at http://journals.plos.org/plosmedicine/s/data-availability and FAQs at http://journals.plos.org/plosmedicine/s/data-availability#loc-faqs-for-data-policy"

* The Data Availability Statement (DAS) requires revision. For each data source used in your study:

FIGURES AND TABLES

* Please provide titles and legends for all figures and tables (including those in Supporting Information files). Please define all acronyms used in each figure or table in its corresponding legend. For Table 2, the legend seems to be part of the table.

* When a p value is given, please specify the statistical test used to determine it in the legend.

CLINICAL TRIALS

* Some of the outcome measures or methods appear to differ between the submitted manuscript and the trial registry and/or protocol. Please clarify and explain the discrepancy. If the outcomes were not prespecified in the protocol, please indicate that they were post hoc and explain why they were added. Post hoc comparisons should be presented as hypothesis generating rather than conclusive.

Comments from Reviewers:

Reviewer #2: I thank the authors for their considered responses to my comments. However, I ask the authors to reconsider two of my previous comments; I do not feel that these have been adequately addressed.

1. I had previously stated that this is an example of a cluster randomised trial (since pregnant women were randomised and outcomes measured on their babies, and mothers had multiple babies) - I stand by this. This is an example of a trial where some clusters were of size 2, and some of size 3; if outcomes were measured only on mothers (i.e. if there were no infant-level outcomes) then I would agree that this is not a cluster randomised trial. However, since the primary outcome is measured on babies, this is classic example of a cluster randomised trial. Please see this paper on accounting for twins in sample size calculations (PMID: 29727020). (This may be useful for any future studies the investigators plan to conduct in a similar population - it indicates how twins may be accounted for). I also note that if the within-mother correlation was not accounted for then this study was likely underpowered to detect the effect of interest (nothing can be done about this now, but it should be noted).

2. I continue to disagree with the per-protocol analysis as described, and continue to recommend alternative approaches (e.g. estimation of complier average causal effects as stated in my previous comment). If the authors do not wish to perform this analysis and instead include the per protocol analysis, the limitations of this must be clearly stated - i.e. the result observed may be due to confounding (those who adhered may be very different from those who did not) and thus is difficult to interpret.

3. Instead of stating "and continuous outcomes were analysed using GEE with a linear distribution and identity link with mean differences and the corresponding 95% confidence intervals along with the p-value." It would be more appropriate to state "and continuous outcomes were analysed using GEE with an identity link with mean differences and the corresponding 95% confidence intervals along with the p-value reported."

Reviewer #4: The authors have extensively addressed all comments raised by the reviewers and editors and have modified the manuscript accordingly. The manuscript is improved with these additions.

[LINK]

---

## [Editor Report · Decision Letter 3]

2 Oct 2025

Dear Dr van Dijk, 

On behalf of my colleagues and the Academic Editor, Andrew Shennan, I am pleased to inform you that we have agreed to publish your manuscript "Cervical pessary versus vaginal progesterone in women with a multiple pregnancy and a short cervix: a randomised controlled trial." (PMEDICINE-D-25-00762R3) in PLOS Medicine.

PRESS

Sincerely, 

Suzanne De Bruijn, PhD 

Associate Editor 

PLOS Medicine